# Self-paced Contrastive Learning with Hybrid Memory for Domain Adaptive Object Re-ID

**Yixiao Ge**    **Feng Zhu**    **Dapeng Chen**    **Rui Zhao**    **Hongsheng Li**

Multimedia Laboratory
The Chinese University of Hong Kong

{yxge@link,hsli@ee}.cuhk.edu.hk    dapengchenxjtu@gmail.com

## Abstract

Domain adaptive object re-ID aims to transfer the learned knowledge from the labeled source domain to the unlabeled target domain to tackle the open-class re-identification problems. Although state-of-the-art pseudo-label-based methods [11, 54, 10, 55, 14] have achieved great success, they did not make full use of all valuable information because of the domain gap and unsatisfying clustering performance. To solve these problems, we propose a novel self-paced contrastive learning framework with hybrid memory. The hybrid memory dynamically generates source-domain class-level, target-domain cluster-level and un-clustered instance-level supervisory signals for learning feature representations. Different from the conventional contrastive learning strategy, the proposed framework jointly distinguishes source-domain classes, and target-domain clusters and un-clustered instances. Most importantly, the proposed self-paced method gradually creates more reliable clusters to refine the hybrid memory and learning targets, and is shown to be the key to our outstanding performance. Our method outperforms state-of-the-arts on multiple domain adaptation tasks of object re-ID and even boosts the performance on the source domain without any extra annotations. Our generalized version on unsupervised object re-ID surpasses state-of-the-art algorithms by considerable **16.7%** and **7.9%** on Market-1501 and MSMT17 benchmarks[†].

## 1 Introduction

Unsupervised domain adaptation (UDA) for object re-identification (re-ID) aims at transferring the learned knowledge from the labeled source domain (dataset) to properly measure the inter-instance affinities in the unlabeled target domain (dataset). Common object re-ID problems include person re-ID and vehicle re-ID, where the source-domain and target-domain data do not share the same identities (classes). Existing UDA methods on object re-ID [38, 11, 54, 10, 55, 45] generally tackled this problem following a two-stage training scheme: (1) supervised pre-training on the source domain, and (2) unsupervised fine-tuning on the target domain. For stage-2 unsupervised fine-tuning, a pseudo-label-based strategy was found effective in state-of-the-art methods [11, 54, 10, 55], which alternates between generating pseudo classes by clustering target-domain instances and training the network with generated pseudo classes. In this way, the source-domain pre-trained network can be adapted to capture the inter-sample relations in the target domain with noisy pseudo-class labels.

Although the pseudo-label-based methods have led to great performance advances, we argue that there exist two major limitations that hinder their further improvements (Figure 1 (a)). (1) During the target-domain fine-tuning, the source-domain images were either not considered [11, 54, 10, 55] or were even found harmful to the final performance [14] because of the limitations of their methodology

---

[*]Dapeng Chen is the corresponding author.
[†]Code is available at https://github.com/yxgee/SpCL.

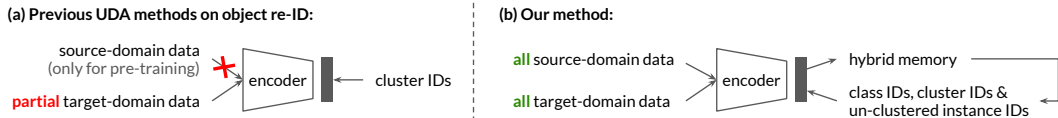

Figure 1: State-of-the-arts [11, 10, 55, 54] on UDA object re-ID discarded both the source-domain data and target-domain un-clustered data for training, while our proposed self-paced contrastive learning framework fully exploits all available data with hybrid memory for joint feature learning.

designs. The accurate source-domain ground-truth labels are valuable but were ignored during target-domain training. (2) Since the clustering process might result in individual outliers, to ensure the reliability of the generated pseudo labels, existing methods [11, 10, 55, 14] simply discarded the outliers from being used for training. However, such outliers might actually be difficult but valuable samples in the target domain and there are generally many outliers especially in early epochs. Simply abandoning them might critically hurt the final performance.

To overcome the problems, we propose a *hybrid memory* to encode all available information from both source and target domains for feature learning. For the source-domain data, their ground-truth class labels can naturally provide valuable supervisions. For the target-domain data, clustering can be conducted to obtain relatively confident clusters as well as un-clustered outliers. All the source-domain class centroids, target-domain cluster centroids, and target-domain un-clustered instance features from the hybrid memory can provide supervisory signals for jointly learning discriminative feature representations across the two domains (Figure 1 (b)). A unified framework is developed for dynamically updating and distinguishing different entries in the proposed hybrid memory.

Specifically, since all the target-domain clusters and un-clustered instances are equally treated as independent classes, the clustering reliability would significantly impact the learned representations. We thus propose a *self-paced contrastive learning* strategy, which initializes the learning process by using the hybrid memory with the most reliable target-domain clusters. Trained with such reliable clusters, the discriminativeness of feature representations can be gradually improved and additional reliable clusters can be formed by incorporating more un-clustered instances into the new clusters. Such a strategy can effectively mitigate the effects of noisy pseudo labels and boost the feature learning process. To properly measure the cluster reliability, a novel multi-scale clustering reliability criterion is proposed, based on which only reliable clusters are preserved and other confusing clusters are disassembled back to un-clustered instances. In this way, our self-paced learning strategy gradually creates more reliable clusters to dynamically refine the hybrid memory and learning targets.

Our contributions are summarized as three-fold. (1) We propose a unified contrastive learning framework to incorporate all available information from both source and target domains for joint feature learning. It dynamically updates the hybrid memory to provide class-level, cluster-level and instance-level supervisions. (2) We design a self-paced contrastive learning strategy with a novel clustering reliability criterion to prevent training error amplification caused by noisy pseudo-class labels. It gradually generates more reliable target-domain clusters for learning better features in the hybrid memory, which in turn, improves clustering. (3) Our method significantly outperforms state-of-the-arts [11, 54, 10, 55, 45] on multiple domain adaptation tasks of object re-ID with up to **5.0%** mAP gains. The proposed unified framework could even boost the performance on the source domain with large margins (**6.6%**) by jointly training with un-annotated target-domain data, while most existing UDA methods "forget" the source domain after fine-tuning on the target domain. Our unsupervised version without labeled source-domain data on object re-ID task significantly outperforms state-of-the-arts [26, 45, 53] by **16.7%** and **7.9%** in terms of mAP on Market-1501 and MSMT17 benchmarks.

## 2 Related Works

**Unsupervised domain adaptation (UDA) for object re-ID.** Existing UDA methods for object re-ID can be divided into two main categories, including pseudo-label-based methods [38, 10, 55, 11, 54, 62, 52, 45] and domain translation-based methods [8, 46, 5, 14]. This paper follows the former one since the pseudo labels were found more effective to capture the target-domain distributions. Though driven by different motivations, previous pseudo-label-based methods generally adopted a two-stage training scheme: (1) pre-training on the source domain with ground-truth IDs, and (2)

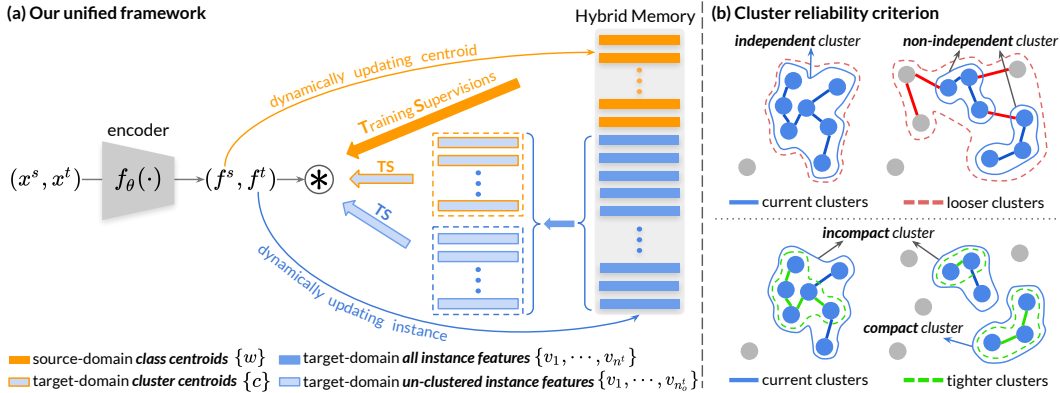

**(a) Our unified framework**

**(b) Cluster reliability criterion**

- source-domain **class centroids** $\{w\}$
- target-domain **cluster centroids** $\{c\}$
- target-domain **all instance features** $\{v_1, \cdots, v_{n^t}\}$
- target-domain **un-clustered instance features** $\{v_1, \cdots, v_{n_u^t}\}$

Figure 2: (a) The illustration of the proposed unified framework with a novel hybrid memory. (b) The proposed reliability criterion for measuring the cluster independence[‡] and compactness.

adapting the target domain with pseudo labels. The pseudo labels can be generated by either clustering instance features [38, 10, 55, 11, 54] or measuring similarities with exemplar features [62, 52, 45], where the clustering-based pipeline maintains state-of-the-art performance to date. The major challenges faced by clustering-based methods is how to improve the precision of pseudo labels and how to mitigate the effects caused by noisy pseudo labels. SSG [10] adopted human local features to assign multi-scale pseudo labels. PAST [55] introduced to utilize multiple regularizations alternately. MMT [11] proposed to generate more robust soft labels via the mutual mean-teaching. AD-Cluster [54] incorporated style-translated images to improve the discriminativeness of instance features. Although various attempts along this direction have led to great performance advances, they ignored to fully exploit all valuable information across the two domains which limits their further improvements, *i.e.*, they simply discarded both the source-domain labeled images and target-domain un-clustered outliers when fine-tuning the model on the target domain with pseudo labels.

**Contrastive learning.** State-of-the-art methods on unsupervised visual representation learning [33, 48, 19, 44, 65, 17, 4] are based on the contrastive learning. Being cast as either the dictionary look-up task [48, 17] or the consistent learning task [44, 4], a contrastive loss was adopted to learn instance discriminative representations by treating each unlabeled sample as a distinct class. Although the instance-level contrastive loss could be used to train embeddings that can be generalized well to downstream tasks with fine-tuning, it does not perform well on the domain adaptive object re-ID tasks which require to correctly measure the inter-class affinities on the unsupervised target domain.

**Self-paced learning.** The "easy-to-hard" training scheme is at the core of self-paced learning [21], which was originally found effective in supervised learning methods, especially with noisy labels [15, 20, 24, 13]. Recently, some methods [41, 16, 6, 56, 67] incorporated the conception of self-paced learning into unsupervised learning tasks by starting the training process with the most confident pseudo labels. However, the self-paced policies designed in these methods were all based on the close-set problems with pre-defined classes, which cannot be generalized to our open-set object re-ID task with completely unknown classes on the target domain. Moreover, they did not consider how to plausibly train with hard samples that cannot be assigned confident pseudo labels all the time.

## 3   Methodology

To tackle the challenges in unsupervised domain adaptation (UDA) on object re-ID, we propose a self-paced contrastive learning framework (Figure 2 (a)), which consists of a CNN [22]-based encoder $f_\theta$ and a novel hybrid memory. The key innovation of the proposed framework lies in jointly training the encoder with all the source-domain class-level, target-domain cluster-level and target-domain un-clustered instance-level supervisions, which are dynamically updated in the hybrid memory to gradually provide more confident learning targets. In order to avoid training error amplification caused by noisy clusters, the self-paced learning strategy initializes the training process with the most

---

[‡]Throughout this paper, the term *independence* is used in its idiomatic sense rather than the statistical sense.

reliable clusters and gradually incorporates more un-clustered instances to form new reliable clusters. A novel reliability criterion is introduced to measure the quality of clusters (Figure 2 (b)).

Our training scheme alternates between two steps: (1) grouping the target-domain samples into clusters and un-clustered instances by clustering the target-domain instance features in the hybrid memory with the self-paced strategy (Section 3.2), and (2) optimizing the encoder $f_\theta$ with a unified contrastive loss and dynamically updating the hybrid memory with encoded features (Section 3.1).

## 3.1 Constructing and Updating Hybrid Memory for Contrastive Learning

Given the target-domain training samples $\mathbb{X}^t$ without any ground-truth label, we employ the self-paced clustering strategy (Section 3.2) to group the samples into clusters and the un-clustered outliers. The whole training set of both domains can therefore be divided into three parts, including the source-domain samples $\mathbb{X}^s$ with ground-truth identity labels, the target-domain pseudo-labeled data $\mathbb{X}_c^t$ within clusters and the target-domain instances $\mathbb{X}_o^t$ not belonging to any cluster, *i.e.*, $\mathbb{X}^t = \mathbb{X}_c^t \cup \mathbb{X}_o^t$. State-of-the-art UDA methods [11, 54, 10, 55] simply abandon all source-domain data and target-domain un-clustered instances, and utilize only the target-domain pseudo labels for adapting the network to the target domain, which, in our opinion, is a sub-optimal solution. Instead, we design a novel contrastive loss to fully exploit available data by treating all the source-domain classes, target-domain clusters and target-domain un-clustered instances as independent classes.

### 3.1.1 Unified Contrastive Learning

Given a general feature vector $\boldsymbol{f} = f_\theta(x), x \in \mathbb{X}^s \cup \mathbb{X}_c^t \cup \mathbb{X}_o^t$, our unified contrastive loss is

$$\mathcal{L}_{\boldsymbol{f}} = -\log \frac{\exp\left(\langle \boldsymbol{f}, \boldsymbol{z}^+ \rangle / \tau\right)}{\sum_{k=1}^{n^s} \exp\left(\langle \boldsymbol{f}, \boldsymbol{w}_k \rangle / \tau\right) + \sum_{k=1}^{n_c^t} \exp\left(\langle \boldsymbol{f}, \boldsymbol{c}_k \rangle / \tau\right) + \sum_{k=1}^{n_o^t} \exp\left(\langle \boldsymbol{f}, \boldsymbol{v}_k \rangle / \tau\right)}, \quad (1)$$

where $\boldsymbol{z}^+$ indicates the positive class prototype corresponding to $\boldsymbol{f}$, the temperature $\tau$ is empirically set as $0.05$ and $\langle \cdot, \cdot \rangle$ denotes the inner product between two feature vectors to measure their similarity. $n^s$ is the number of source-domain classes, $n_c^t$ is the number of target-domain clusters and $n_o^t$ is the number of target-domain un-clustered instances. More specifically, if $\boldsymbol{f}$ is a source-domain feature, $\boldsymbol{z}^+ = \boldsymbol{w}_k$ is the centroid of the source-domain class $k$ that $\boldsymbol{f}$ belongs to. If $\boldsymbol{f}$ belongs to the $k$-th target-domain cluster, $\boldsymbol{z}^+ = \boldsymbol{c}_k$ is the $k$-th cluster centroid. If $\boldsymbol{f}$ is a target-domain un-clustered outlier, we would have $\boldsymbol{z}^+ = \boldsymbol{v}_k$ as the outlier instance feature corresponding to $\boldsymbol{f}$. Intuitively, the above joint contrastive loss encourages the encoded feature vector to approach its assigned classes, clusters or instances. Note that we utilize class centroids $\{\boldsymbol{w}\}$ instead of learnable class weights for encoding source-domain classes to match their semantics to those of the clusters' or outliers' centroids. Our experiments (Section 4.4) show that, if the semantics of class-level, cluster-level and instance-level supervisions do not match, the performance drops significantly.

**Discussion.** The most significant difference between our unified contrastive loss (Eq. (1)) and previous contrastive losses [48, 17, 4, 33] is that ours jointly distinguishes classes, clusters, and un-clustered instances, while previous ones only focus on separating instances without considering any ground-truth classes or pseudo-class labels as our method does. They target at instance discrimination task but fail in properly modeling intra-/inter-class affinities on domain adaptive re-ID tasks.

### 3.1.2 Hybrid Memory

As the cluster number $n_c^t$ and outlier instance number $n_o^t$ may change during training with the alternate clustering strategy, the class prototypes for the unified contrastive loss (Eq. (1)) are built in a non-parametric and dynamic manner. We propose a novel hybrid memory to provide the source-domain class centroids $\{\boldsymbol{w}_1, \cdots, \boldsymbol{w}_{n^s}\}$, target-domain cluster centroids $\{\boldsymbol{c}_1, \cdots, \boldsymbol{c}_{n_c^t}\}$ and target-domain un-clustered instance features $\{\boldsymbol{v}_1, \cdots, \boldsymbol{v}_{n_o^t}\}$. For continuously storing and updating the above three types of entries, we propose to cache source-domain *class* centroids $\{\boldsymbol{w}_1, \cdots, \boldsymbol{w}_{n^s}\}$ and all the target-domain *instance* features $\{\boldsymbol{v}_1, \cdots, \boldsymbol{v}_{n^t}\}$ simultaneously in the hybrid memory, where $n^t$ is the number of all the target-domain instances and $n^t \neq n_c^t + n_o^t$. Without loss of generality, we assume that un-clustered features in $\{\boldsymbol{v}\}$ have indices $\{1, \cdots, n_o^t\}$, while other clustered features in $\{\boldsymbol{v}\}$ have indices from $n_o^t + 1$ to $n^t$. In other words, $\{\boldsymbol{v}_{n_o^t+1}, \cdots, \boldsymbol{v}_{n^t}\}$ dynamically form the cluster centroids $\{\boldsymbol{c}\}$ while $\{\boldsymbol{v}_1, \cdots, \boldsymbol{v}_{n_o^t}\}$ remain un-clustered instances.

**Memory initialization.** The hybrid memory is initialized with the extracted features by performing forward computation of $f_\theta$: the initial source-domain class centroids $\{w\}$ can be obtained as the mean feature vectors of each class, while the initial target-domain instance features $\{v\}$ are directly encoded by $f_\theta$. After that, the target-domain cluster centroids $\{c\}$ are initialized with the mean feature vectors of each cluster from $\{v\}$, *i.e.*,

$$c_k = \frac{1}{|\mathcal{I}_k|} \sum_{v_i \in \mathcal{I}_k} v_i, \tag{2}$$

where $\mathcal{I}_k$ denotes the $k$-th cluster set that contains all the feature vectors within cluster $k$ and $|\cdot|$ denotes the number of features in the set. Note that the source-domain class centroids $\{w\}$ and the target-domain instance features $\{v\}$ are only initialized once by performing the forward computation at the beginning of the learning algorithm, and then can be continuously updated during training.

**Memory update.** At each iteration, the encoded feature vectors in each mini-batch would be involved in hybrid memory updating. For the source-domain class centroids $\{w\}$, the $k$-th centroid $w_k$ is updated by the mean of the encoded features belonging to class $k$ in the mini-batch as

$$w_k \leftarrow m^s w_k + (1 - m^s) \cdot \frac{1}{|\mathcal{B}_k|} \sum_{f_i^s \in \mathcal{B}_k} f_i^s, \tag{3}$$

where $\mathcal{B}_k$ denotes the feature set belonging to source-domain class $k$ in the current mini-batch and $m^s \in [0, 1]$ is a momentum coefficient for updating source-domain class centroids. $m^s$ is empirically set as $0.2$.

The target-domain cluster centroids cannot be stored and updated in the same way as the source-domain class centroids, since the clustered set $\mathbb{X}_c^t$ and un-clustered set $\mathbb{X}_o^t$ are constantly changing. As the hybrid memory caches all the target-domain features $\{v\}$, each encoded feature vector $f_i^t$ in the mini-batch is utilized to update its corresponding instance entry $v_i$ by

$$v_i \leftarrow m^t v_i + (1 - m^t) f_i^t, \tag{4}$$

where $m^t \in [0, 1]$ is the momentum coefficient for update target-domain instance features and is set as $0.2$ in our experiments. Given the updated instance memory $v_i$, if $f_i^t$ belongs to the cluster $k$, the corresponding centroid $c_k$ needs to be updated with Eq. (2).

**Discussion.** The hybrid memory has two main differences from the memory used in [48, 17]: (1) Our hybrid memory caches prototypes for both the centroids and instances, while the memory in [48, 17] only provides instance-level prototypes. Other than the centroids, we for the first time treat clusters and instances as equal classes; (2) The cluster/instance learning targets provided by our hybrid memory are gradually updated and refined, while previous memory [48, 17] only supports fixed instance-level targets. Note that our self-paced strategy (will be discussed in Section 3.2) dynamically determines confident clusters and un-clustered instances.

The momentum updating strategy is inspired by [17, 43], and we further introduce how to update hybrid prototypes, *i.e.*, centroids and instances. Note that we employ different updating strategies for class centroids (Eq. (3)) and cluster centroids (Eq. (4)&(2)) since source-domain classes are fixed while target-domain clusters are dynamically changed.

## 3.2 Self-paced Learning with Reliable Clusters

A simple way to split the target-domain data into clusters $\mathbb{X}_c^t$ and un-clustered outliers $\mathbb{X}_o^t$ is to cluster the target-domain instance features $\{v_1, \cdots, v_{n^t}\}$ from the hybrid memory by a certain algorithm (*e.g.*, DBSCAN [9]). Since all the target-domain clusters and un-clustered outlier instances are treated as distinct classes in Eq. (1), the clustering reliability would significantly impact the learned representations. If the clustering is perfect, merging all the instances into their true clusters would no doubt improve the final performance (denotes as "oracle" in Table 5). However, in practice, merging an instance into a wrong cluster does more harm than good. A self-paced learning strategy is therefore introduced, where in the re-clustering step before each epoch, only the most reliable clusters are preserved and the unreliable clusters are disassembled back to un-clustered instances. A reliability criterion is proposed to identify unreliable clusters by measuring the independence and compactness.

**Independence of clusters.** A reliable cluster should be independent from other clusters and individual samples. Intuitively, if a cluster is far away from other samples, it can be considered as highly independent. However, due to the uneven density in the latent space, we cannot naïvely use the distances between the cluster centroid and outside-cluster samples to measure the cluster independence. Generally, the clustering results can be tuned by altering certain hyper-parameters of the clustering criterion. One can *loosen* the clustering criterion to possibly include *more* samples in each cluster or *tighten* the clustering criterion to possibly include *fewer* samples in each cluster. We denote the samples within the same cluster of $\boldsymbol{f}_i^t$ as $\mathcal{I}(\boldsymbol{f}_i^t)$. We propose the following metric to measure the cluster independence, which is formulated as an intersection-over-union (IoU) score,

$$\mathcal{R}_{\text{indep}}(\boldsymbol{f}_i^t) = \frac{|\mathcal{I}(\boldsymbol{f}_i^t) \cap \mathcal{I}_{\text{loose}}(\boldsymbol{f}_i^t)|}{|\mathcal{I}(\boldsymbol{f}_i^t) \cup \mathcal{I}_{\text{loose}}(\boldsymbol{f}_i^t)|} \in [0, 1], \tag{5}$$

where $\mathcal{I}_{\text{loose}}(\boldsymbol{f}_i^t)$ is the cluster set containing $\boldsymbol{f}_i^t$ when the clustering criterion becomes looser. Larger $\mathcal{R}_{\text{indep}}(\boldsymbol{f}_i^t)$ indicates a more independent cluster for $\boldsymbol{f}_i^t$, *i.e.*, even one looses the clustering criterion, there would be no more sample to be included into the new cluster $\mathcal{I}_{\text{loose}}(\boldsymbol{f}_i^t)$. Samples within the same cluster set (*e.g.*, $\mathcal{I}(\boldsymbol{f}_i^t)$) generally have the same independence score.

**Compactness of clusters.** A reliable cluster should also be compact, *i.e.*, the samples within the same cluster should have small inter-sample distances. In an extreme case, when a cluster is most compact, all the samples in the cluster have zero inter-sample distances. Its samples would not be split into different clusters even when the clustering criterion is tightened. Based on this assumption, we can define the following metric to determine the compactness of the clustered point $\boldsymbol{f}_i^t$ as

$$\mathcal{R}_{\text{comp}}(\boldsymbol{f}_i^t) = \frac{|\mathcal{I}(\boldsymbol{f}_i^t) \cap \mathcal{I}_{\text{tight}}(\boldsymbol{f}_i^t)|}{|\mathcal{I}(\boldsymbol{f}_i^t) \cup \mathcal{I}_{\text{tight}}(\boldsymbol{f}_i^t)|} \in [0, 1], \tag{6}$$

where $\mathcal{I}_{\text{tight}}(\boldsymbol{f}_i^t)$ is the cluster set containing $\boldsymbol{f}_i^t$ when tightening the criterion. Larger $\mathcal{R}_{\text{comp}}(\boldsymbol{f}_i^t)$ indicates smaller inter-sample distances around $\boldsymbol{f}_i^t$ within $\mathcal{I}(\boldsymbol{f}_i^t)$, since a cluster with larger inter-sample distances is more likely to include fewer points when a tightened criterion is adopted. The same cluster's data points may have different compactness scores due to the uneven density.

Given the above metrics for measuring the cluster reliability, we could compute the independence and compactness scores for each data point within clusters. We set up $\alpha, \beta \in [0, 1]$ as independence and compactness thresholds for determining reliable clusters. Specifically, we preserve independent clusters with compact data points whose $\mathcal{R}_{\text{indep}} > \alpha$ and $\mathcal{R}_{\text{comp}} > \beta$, while the remaining data are treated as un-clustered outlier instances. With the update of the encoder $f_\theta$ and target-domain instance features $\{\boldsymbol{v}\}$ from the hybrid memory, more reliable clusters can be gradually created to further improve the feature learning. The overall algorithm is detailed in Alg. 1 of Appendix A.

## 4 Experiments

### 4.1 Datasets and Evaluation Protocol

Table 1: Statistics of the datasets used for training and evaluation. (*) denotes the synthetic datasets.

| Dataset | # train IDs | # train images | # test IDs | # query images | # cameras | # total images |
|---|---|---|---|---|---|---|
| Market-1501 [58] | 751 | 12,936 | 750 | 3,368 | 6 | 32,217 |
| MSMT17 [46] | 1,041 | 32,621 | 3,060 | 11,659 | 15 | 126,441 |
| PersonX [39]* | 410 | 9,840 | 856 | 5,136 | 6 | 45,792 |
| VeRi-776 [28] | 575 | 37,746 | 200 | 1,678 | 20 | 51,003 |
| VehicleID [27] | 13,164 | 113,346 | 800 | 5,693 | - | 221,763 |
| VehicleX [32]* | 1,362 | 192,150 | - | - | 11 | 192,150 |

We evaluate our proposed method on both the mainstream real→real adaptation tasks and the more challenging synthetic→real adaptation tasks in person re-ID and vehicle re-ID problems. As shown in Table 1, two real-world person datasets and one synthetic person dataset, as well as two real-world vehicle datasets and one synthetic vehicle dataset, are adopted in our experiments.

**Person re-ID datasets[†].** The Market-1501 and MSMT17 are widely used real-world person image datasets in domain adaptive tasks, among which, MSMT17 has the most images and is most challenging. The synthetic PersonX [39] is generated based on Unity [36] with manually designed obstacles, *e.g.*, random occlusion, resolution and illumination differences, *etc.*

---

[†]DukeMTMC-reID [37] dataset has been taken down and should no longer be used.

Table 2: Comparison with state-of-the-art methods on unsupervised domain adaptation for object re-ID. (*) the implementation is based on the authors' code.

(a) *Real→real* adaptation on person re-ID datasets.

| Methods | | Market-1501→MSMT17 | | | |
|---|---|---|---|---|---|
| | | mAP | top-1 | top-5 | top-10 |
| PTGAN [46] | CVPR'18 | 2.9 | 10.2 | - | 24.4 |
| ECN [62] | CVPR'19 | 8.5 | 25.3 | 36.3 | 42.1 |
| SSG [10] | ICCV'19 | 13.2 | 31.6 | - | 49.6 |
| ECN++ [63] | TPAMI'20 | 15.2 | 40.4 | 53.1 | 58.7 |
| MMT-$k$means [11] | ICLR'20 | 22.9 | 49.2 | 63.1 | 68.8 |
| MMCL [45] | CVPR'20 | 15.1 | 40.8 | 51.8 | 56.7 |
| DG-Net++ [66] | ECCV'20 | 22.1 | 48.4 | 60.9 | 66.1 |
| D-MMD [31] | ECCV'20 | 13.5 | 29.1 | 46.3 | 54.1 |
| JVTC [23] | ECCV'20 | 20.3 | 45.4 | 58.4 | 64.3 |
| GPR [29] | ECCV'20 | 20.4 | 43.7 | 56.1 | 61.9 |
| NRMT [57] | ECCV'20 | 19.8 | 43.7 | 56.5 | 62.2 |
| MMT-dbscan [11]* | ICLR'20 | 24.0 | 50.1 | 63.5 | 69.3 |
| **Ours** | | **26.8** | **53.7** | **65.0** | **69.8** |
| Methods | | MSMT17→Market-1501 | | | |
| | | mAP | top-1 | top-5 | top-10 |
| MAR [52] | CVPR'19 | 40.0 | 67.7 | 81.9 | - |
| PAUL [50] | CVPR'19 | 40.1 | 68.5 | 82.4 | 87.4 |
| CASCL [47] | ICCV'19 | 35.5 | 65.4 | 80.6 | 86.2 |
| DG-Net++ [66] | ECCV'20 | 64.6 | 83.1 | 91.5 | 94.3 |
| D-MMD [31] | ECCV'20 | 50.8 | 72.8 | 88.1 | 92.3 |
| MMT-dbscan [11]* | ICLR'20 | 75.6 | 89.3 | 95.8 | 97.5 |
| **Ours** | | **77.5** | **89.7** | **96.1** | **97.6** |

(b) *Synthetic→real* adaptation on person re-ID datasets.

| Methods | | PersonX→MSMT17 | | | |
|---|---|---|---|---|---|
| | | mAP | top-1 | top-5 | top-10 |
| MMT-dbscan [11]* | ICLR'20 | 17.7 | 39.1 | 52.6 | 58.5 |
| **Ours** | | **22.7** | **47.7** | **60.0** | **65.5** |
| Methods | | PersonX→Market-1501 | | | |
| | | mAP | top-1 | top-5 | top-10 |
| MMT-dbscan [11]* | ICLR'20 | 71.0 | 86.5 | 94.8 | **97.0** |
| **Ours** | | **73.8** | **88.0** | **95.3** | 96.9 |

(c) *Real→real* and *synthetic→real* adaptation on vehicle re-ID datasets.

| Methods | | VehicleID→VeRi-776 | | | |
|---|---|---|---|---|---|
| | | mAP | top-1 | top-5 | top-10 |
| MMT-dbscan [11]* | ICLR'20 | 35.3 | 74.6 | 82.6 | 87.0 |
| **Ours** | | **38.9** | **80.4** | **86.8** | **89.6** |
| Methods | | VehicleX→VeRi-776 | | | |
| | | mAP | top-1 | top-5 | top-10 |
| MMT-dbscan [11]* | ICLR'20 | 35.6 | 76.0 | 83.1 | 87.4 |
| **Ours** | | **38.9** | **81.3** | **87.3** | **90.0** |

Table 3: Comparison with state-of-the-art UDA methods and supervised learning methods when evaluating on the labeled source domain. (*) the implementation is based on the authors' code.

| Methods | | MSMT17→Market-1501 | | | | Market-1501→MSMT17 | | | |
|---|---|---|---|---|---|---|---|---|---|
| | | mAP | top-1 | top-5 | top-10 | mAP | top-1 | top-5 | top-10 |
| MMT-dbscan [11]* | ICLR'20 | 3.2 | 9.8 | 16.8 | 20.7 | 30.7 | 59.9 | 75.7 | 81.3 |
| Encoder train/test on the source domain | | 49.9 | 75.2 | 86.4 | 89.8 | 84.7 | 94.0 | 97.7 | 98.7 |
| **Ours test on the source domain** | | **56.5 (+6.6)** | **79.4** | **88.7** | **91.3** | **86.8 (+2.1)** | **94.7** | **97.9** | **98.6** |
| *Supervised learning methods on source* | | MSMT17 | | | | Market-1501 | | | |
| FD-GAN [12] | NeurIPS'18 | - | - | - | - | 77.7 | 90.5 | - | - |
| DG-Net [59] | CVPR'19 | 52.3 | 77.2 | 87.4 | 90.5 | 86.0 | 94.8 | - | - |
| OSNet [64] | ICCV'19 | 52.9 | 78.7 | - | - | 84.9 | 94.8 | - | - |
| Circle loss [40] | CVPR'20 | 50.2 | 76.3 | - | - | 84.9 | 94.2 | - | - |

**Vehicle re-ID datasets.** Although domain adaptive person re-ID has been long studied, the same task on the vehicle has not been fully explored. We conduct experiments with the real-world VeRi-776, VehicleID and the synthetic VehicleX datasets. VehicleX [32] is also generated by the Unity engine [51, 42] and further translated to have the real-world style by SPGAN [8].

**Evaluation protocol.** In the experiments, only ground-truth IDs on the source-domain datasets are provided for training. Mean average precision (mAP) and cumulative matching characteristic (CMC), proposed in [58], are adopted to evaluate the methods' performances on the target-domain datasets. No post-processing technique, *e.g.*, re-ranking [60] or multi-query fusion [58], is adopted.

## 4.2 Implementation Details

We adopt an ImageNet-pretrained [7] ResNet-50 [18] as the backbone for the encoder $f_\theta$. Following the clustering-based UDA methods [11, 10, 38], we use DBSCAN [9] for clustering before each epoch. The maximum distance between neighbor points, which is the most important parameter in DBSCAN, is tuned to loosen or tighten the clustering in our proposed self-paced learning strategy. We use a constant threshold $\alpha$ and dynamic threshold $\beta$ for identifying independent clusters with the most compact points by the reliability criterion. More details can be found in Appendix C.

## 4.3 Comparison with State-of-the-arts

**UDA performance on the target domain.** We compare our proposed framework with state-of-the-art UDA methods on multiple domain adaptation tasks in Table 2, including three real→real and three synthetic→real tasks. The tasks in Tables 2b & 2c were not surveyed by previous methods, so we implement state-of-the-art MMT [11] on these datasets for comparison. Our method significantly outperforms all state-of-the-arts on both person and vehicle datasets with a plain ResNet-50 backbone, achieving 2-4% improvements in terms of mAP on the common real→real tasks and up to 5.0% increases on the challenging synthetic→real tasks. An inspiring discovery is that the synthetic→real task could achieve competitive performance as the real→real task with the same target-domain dataset

Table 4: Comparison with state-of-the-art methods on the unsupervised object re-ID task without the labeled source-domain data. (*) the implementation is based on the authors' code.

| Methods | | Market-1501 | | | |
|---|---|---|---|---|---|
| | | mAP | top-1 | top-5 | top-10 |
| OIM [49] | CVPR'17 | 14.0 | 38.0 | 58.0 | 66.3 |
| BUC [25] | AAAI'19 | 38.3 | 66.2 | 79.6 | 84.5 |
| SSL [26] | CVPR'20 | 37.8 | 71.7 | 83.8 | 87.4 |
| MMCL [45] | CVPR'20 | 45.5 | 80.3 | 89.4 | 92.3 |
| HCT [53] | CVPR'20 | 56.4 | 80.0 | 91.6 | 95.2 |
| MoCo [17]* | CVPR'20 | 6.1 | 12.8 | 27.1 | 35.7 |
| **Ours w/o source data** | | **73.1** | **88.1** | **95.1** | **97.0** |

| Methods | | MSMT17 | | | |
|---|---|---|---|---|---|
| | | mAP | top-1 | top-5 | top-10 |
| MMCL [45] | CVPR'20 | 11.2 | 35.4 | 44.8 | 49.8 |
| MoCo [17]* | CVPR'20 | 1.6 | 4.3 | 9.7 | 13.5 |
| **Ours w/o source data** | | **19.1** | **42.3** | **55.6** | **61.2** |

| Methods | | VeRi-776 | | | |
|---|---|---|---|---|---|
| | | mAP | top-1 | top-5 | top-10 |
| MoCo [17]* | CVPR'20 | 9.5 | 24.9 | 40.6 | 51.8 |
| **Ours w/o source data** | | **36.9** | **79.9** | **86.8** | **89.9** |

Table 5: Ablation studies of our proposed self-paced contrastive learning on individual components.

(a) Experiments on domain adaptive person re-ID.

| Methods | MSMT→Market-1501 | | | |
|---|---|---|---|---|
| | mAP | top-1 | top-5 | top-10 |
| *analysis of the unified contrastive learning mechanism:* | | | | |
| *Src.* class | 25.3 | 51.3 | 69.6 | 76.6 |
| *Src.* class + *tgt.* instance | 4.9 | 12.6 | 24.8 | 32.7 |
| *Src.* class + *tgt.* cluster (*w/o* self-paced) | 28.9 | 50.1 | 64.5 | 71.0 |
| *Src.* class + *tgt.* cluster (*w/* self-paced) | 69.2 | 84.9 | 94.0 | 96.4 |
| *Src.* class → *Src.* learnable weights | 70.7 | 86.9 | 94.1 | 96.3 |
| Ours *w/o* unified contrast | 68.8 | 84.6 | 94.1 | 96.2 |
| *analysis of the self-paced learning strategy:* | | | | |
| Ours *w/o* $\mathcal{R}_{comp}$ & $\mathcal{R}_{indep}$ | 74.5 | 89.1 | 95.3 | 96.8 |
| Ours *w/o* $\mathcal{R}_{comp}$ | 75.4 | 89.3 | 95.5 | 97.1 |
| Ours *w/o* $\mathcal{R}_{indep}$ | 76.6 | 89.5 | 95.6 | 97.3 |
| Oracle | 83.5 | 93.1 | 97.7 | 98.6 |
| **Ours (full)** | **77.5** | **89.7** | **96.1** | **97.6** |

(b) Experiments on unsupervised person re-ID.

| Methods | Market-1501 | | | |
|---|---|---|---|---|
| | mAP | top-1 | top-5 | top-10 |
| *analysis of the unified contrastive learning mechanism:* | | | | |
| *tgt.* instance | 3.5 | 9.1 | 18.7 | 25.8 |
| *tgt.* cluster (*w/o* self-paced) | 6.7 | 16.5 | 27.9 | 33.8 |
| *tgt.* cluster (*w/* self-paced) | 10.1 | 23.9 | 37.3 | 43.2 |
| Ours *w/o* unified contrast | 57.0 | 76.2 | 89.7 | 93.0 |
| *analysis of the self-paced learning strategy:* | | | | |
| Ours *w/o* $\mathcal{R}_{comp}$ & $\mathcal{R}_{indep}$ | 68.2 | 85.0 | 93.3 | 95.3 |
| Ours *w/o* $\mathcal{R}_{comp}$ | 68.6 | 86.2 | 93.9 | 95.8 |
| Ours *w/o* $\mathcal{R}_{indep}$ | 71.8 | 87.5 | 95.0 | 96.8 |
| Oracle | 82.3 | 92.6 | 97.2 | 98.4 |
| **Ours (full) *w/o* source data** | **73.1** | **88.1** | **95.1** | **97.0** |

(*e.g.*, VeRi-776), which indicates that we are one more step closer towards no longer needing any manually annotated real-world images in the future.

**Further improvements on the source domain.** State-of-the-art UDA methods inevitably forget the source-domain knowledge after fine-tuning the pretrained networks on the target domain, as demonstrated by MMT [11] in Table 3. In contrast, our proposed unified framework could effectively model complex inter-sample relations across the two domains, boosting the source-domain performance by up to 6.6% mAP. Note that experiments of "Encoder train/test on the source domain" adopt the same training objective (Eq. (1)) as our proposed method, except for that only source-domain class centroids $\{w\}$ are available. Our method also outperforms state-of-the-art supervised re-ID methods [12, 59, 64, 40] on the source domain without either using multiple losses or more complex networks. Such a phenomenon indicates that our method could be applied to improve the supervised training by incorporating unlabeled data without extra human labor.

**Unsupervised re-ID without any labeled training data.** Another stream of research focuses on training the re-ID model without any labeled data, *i.e.*, excluding source-domain data from the training set. Our method can be easily generalized to such a setting by discarding the source-domain class centroids $\{w\}$ from both the hybrid memory and training objective (See Alg. 2 in Appendix A for details). As shown in Table 4, our method considerably outperforms state-of-the-arts by up to 16.7% improvements in terms of mAP. We also implement state-of-the-art unsupervised method MoCo [17], which adopts the conventional contrastive loss, and unfortunately, it is inapplicable on unsupervised re-ID tasks. MoCo [17] underperforms because it treats each instance as a single class, while the core of re-ID tasks is to encode and model intra-/inter-class variations. MoCo [17] is good at unsupervised pre-training but its resulting networks need finetuning with (pseudo) class labels.

## 4.4 Ablation Studies

We analyse the effectiveness of our proposed unified contrastive loss with hybrid memory and self-paced learning strategy in Table 5. The "oracle" experiment adopts the target-domain ground-truth IDs as cluster labels for training, reflecting the maximal performance with our pipeline.

**Unified contrastive learning mechanism.** In order to verify the necessity of each type of classes in the unified contrastive loss (Eq. (1)), we conduct experiments when removing any one of the source-domain class-level, target-domain cluster-level or un-clustered instance-level supervisions (Table 5a). Baseline "*Src.* class" adopts only source-domain images with ground-truth IDs for training. "*Src.* class + *tgt.* instance" treats each target-domain sample as a distinct class. It totally fails with even worse results than the baseline "*Src.* class", showing that directly generalizing conventional

contrastive loss to UDA tasks is inapplicable. "*Src.* class + *tgt.* cluster" follows existing UDA methods [11, 10, 55, 14], by simply discarding un-clustered instances from training. Noticeable performance drops are observed, especially without the self-paced policy to constrain reliable clusters. Note that the only difference between "*Src.* class + *tgt.* cluster (*w/* self-paced)" and "Ours (full)" is whether using outliers for training and the large performance gaps are due to the facts that: 1) There are many un-clustered outliers ($>$ half of all samples), especially in early epochs; 2) Outliers serve as difficult samples and excluding them over-simplifies the training task; 3) "*Src.* class + *tgt.* cluster" doesn't update outliers in the memory, making them unsuitable to be clustered in the later epochs.

As illustrated in Table 5b, we further verify the necessity of unified training in unsupervised object re-ID tasks. We observe the same trend as domain adaptive tasks: solving the problem via instance discrimination ("*tgt.* instance") would fail. What is different is that, even with our self-paced strategy, training with clusters alone ("*tgt.* cluster") would fail. That is due to the fact that only a few samples take part in the training if discarding the outliers, undoubtedly leading to training collapse. Note that previous unsupervised re-ID methods [25, 53] which abandoned outliers did not fail, since they did not utilize a memory bank that requires all the entries to be continuously updated.

We adopt the non-parametric class centroids to supervise the source-domain feature learning, however, conventional methods generally adopt a learnable classifier for supervised learning. "*Src.* class $\rightarrow$ *Src.* learnable weights" in Table 5a is therefore conducted to verify the necessity of using source-domain class centroids for training to match the semantics of target-domain training supervisions. We also test the effect of not extending negative classes across different types of contrasts. For instance, source-domain samples only treat non-corresponding source-domain classes as their negative classes. "Ours *w/o* unified contrast" shows inferior performance in both Table 5a and 5b. This indicates the effectiveness of the unified contrastive learning between all types of classes in Eq. (1).

**Self-paced learning strategy.** We propose the self-paced learning strategy to preserve the most reliable clusters for providing stronger supervisions. The intuition is to measure the stability of clusters by hierarchical structures, *i.e.*, a reliable cluster should be consistent in clusters at multiple levels. $\mathcal{R}_{\text{indep}}$ and $\mathcal{R}_{\text{comp}}$ are therefore proposed to measure the independence and compactness of clusters, respectively. To verify the effectiveness of such a strategy, we evaluate our framework when removing either $\mathcal{R}_{\text{indep}}$ or $\mathcal{R}_{\text{comp}}$, or both of them. Obvious performance drops are observed under all these settings, *e.g.*, 4.9% mAP drops are shown when removing $\mathcal{R}_{\text{indep}}\&\mathcal{R}_{\text{comp}}$ in Table 5b.

We illustrate the number of clusters and their corresponding Normalized Mutual Information (NMI) scores during training on MSMT17$\rightarrow$Market-1501 in Figure 3. It can be observed that the quantity and quality of clusters are closer to the ground-truth IDs with the proposed self-paced learning strategy regardless of the un-clustered instance-level contrast, indicating higher reliability of the clusters and the effectiveness of the self-paced strategy.

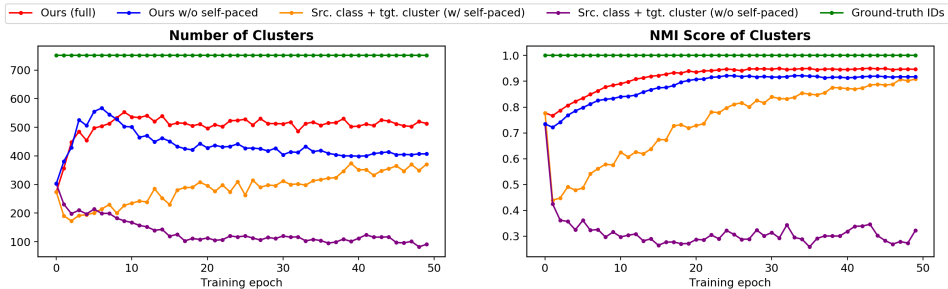

Figure 3: Ablation study by observing the dynamically changing cluster numbers and their corresponding Normalized Mutual Information (NMI) scores during training on MSMT17$\rightarrow$Market-1501.

## 5 Discussion and Conclusion

Our method has shown considerable improvements over a variety of unsupervised or domain adaptive object re-ID tasks. The supervised performance can also be promoted labor-free by incorporating unlabeled data for training in our framework. The core is at exploiting all available data for jointly training with hybrid supervision. Positive as the results are, there still exists a gap from the oracle, suggesting that the pseudo-class labels may not be satisfactory enough even with the proposed self-paced strategy. Further studies are called for. Beyond the object re-ID task, our method has great potential on other unsupervised learning tasks, which needs to be explored.

## Broader Impact

Our method can help to identify and track different types of objects (*e.g.*, vehicles, cyclists, pedestrians, *etc.* ) across different cameras (domains), thus boosting the development of smart retail, smart transportation, and smart security systems in the future metropolises. In addition, our proposed self-paced contrastive learning is quite general and not limited to the specific research field of object re-ID. It can be well extended to broader research areas, including unsupervised and semi-supervised representation learning.

However, object re-ID systems, when applied to identify pedestrians and vehicles in surveillance systems, might give rise to the infringement of people's privacy, since such re-ID systems often rely on non-consensual surveillance data for training, *i.e.*, it is unlikely that all human subjects even knew they were being recorded. Therefore, governments and officials need to carefully establish strict regulations and laws to control the usage of re-ID technologies. Otherwise, re-ID technologies can potentially equip malicious actors with the ability to surveil pedestrians or vehicles through multiple CCTV cameras without their consent. The research committee should also avoid using the datasets with ethics issues, *e.g.*, DukeMTMC [37], which has been taken down due to the violation of data collection terms, should no longer be used. We would not evaluate our method on DukeMTMC related benchmarks as well. Furthermore, we should be cautious of the misidentification of the re-ID systems to avoid possible disturbance. Also, note that the demographic makeup of the datasets used is not representative of the broader population.

## Acknowledgements

This work is supported in part by the General Research Fund through the Research Grants Council of Hong Kong under Grants (*Nos.* CUHK14208417, CUHK14207319), in part by the Hong Kong Innovation and Technology Support Program (*No.* ITS/312/18FX), in part by CUHK Strategic Fund.

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
