[Supplementary Material]

# Self-paced Contrastive Learning with Hybrid Memory for Domain Adaptive Object Re-ID
## Supplementary Material

**Yixiao Ge    Feng Zhu    Dapeng Chen    Rui Zhao    Hongsheng Li**

Multimedia Laboratory
The Chinese University of Hong Kong

{yxge@link,hsli@ee}.cuhk.edu.hk    dapengchenxjtu@gmail.com

## A    Algorithm Details

---

**Algorithm 1** Self-paced contrastive learning algorithm on domain adaptive object re-ID

---

**Require:** Source-domain labeled data $\mathbb{X}^s$ and target-domain unlabeled data $\mathbb{X}^t$;
**Require:** Initialize the backbone encoder $f_\theta$ with ImageNet-pretrained ResNet-50;
**Require:** Initialize the hybrid memory with features extracted by $f_\theta$;
**Require:** Temperature $\tau$ for Eq. (1), momentum $m^s$ for Eq. (3), momentum $m^t$ for Eq. (4);
  **for** n in [1, num_epochs] **do**
    Group $\mathbb{X}^t$ into $\mathbb{X}_c^t$ and $\mathbb{X}_o^t$ by clustering $\{\boldsymbol{v}\}$ from the hybrid memory with the independence Eq. (5) and compactness Eq. (6) criterion;
    Initialize the cluster centroids $\{\boldsymbol{c}\}$ with Eq. (2) in the hybrid memory;
    **for** each mini-batch $\{x_i^s\} \subset \mathbb{X}^s, \{x_i^t\} \subset \mathbb{X}^t$ **do**
      **1:** Encode features $\{\boldsymbol{f}_i^s\}$, $\{\boldsymbol{f}_i^t\}$ for $\{x_i^s\}$, $\{x_i^t\}$ with $f_\theta$;
      **2:** Compute the unified contrastive loss with $\{\boldsymbol{f}_i^s\}$, $\{\boldsymbol{f}_i^t\}$ by Eq. (1) and update the encoder $f_\theta$ by back-propagation;
      **3:** Update source-domain related class centroids $\{\boldsymbol{w}\}$ in the hybrid memory with $\{\boldsymbol{f}_i^s\}$ and momentum $m^s$ (Eq. (3));
      **4:** Update target-domain related instance features $\{\boldsymbol{v}\}$ in the hybrid memory with $\{\boldsymbol{f}_i^t\}$ and momentum $m^t$ (Eq. (4));
      **5:** Update target-domain related cluster centroids $\{\boldsymbol{c}\}$ with updated $\{\boldsymbol{v}\}$ in the hybrid memory (Eq. (2));
    **end for**
  **end for**

---

**Algorithm 2** Self-paced contrastive learning algorithm on unsupervised object re-ID

---

**Require:** Unlabeled data $\mathbb{X}^t$;
**Require:** Initialize the backbone encoder $f_\theta$ with ImageNet-pretrained ResNet-50;
**Require:** Initialize the hybrid memory with features extracted by $f_\theta$;
**Require:** Temperature $\tau$ for Eq. (1), momentum $m^t$ for Eq. (4);
  **for** n in [1, num_epochs] **do**
    Group $\mathbb{X}^t$ into $\mathbb{X}_c^t$ and $\mathbb{X}_o^t$ by clustering $\{\boldsymbol{v}\}$ from the hybrid memory with the independence Eq. (5) and compactness Eq. (6) criterion;
    Initialize the cluster centroids $\{\boldsymbol{c}\}$ with Eq. (2) in the hybrid memory;
    **for** each mini-batch $\{x_i^t\} \subset \mathbb{X}^t$ **do**
      **1:** Encode features $\{\boldsymbol{f}_i^t\}$ for $\{x_i^t\}$ with $f_\theta$;
      **2:** Compute the unsupervised-version unified contrastive loss with $\{\boldsymbol{f}_i^t\}$ as below and update the encoder $f_\theta$ by back-propagation;

$$\mathcal{L}_{\boldsymbol{f}} = -\log \frac{\exp{(\langle \boldsymbol{f}, \boldsymbol{z}^+ \rangle / \tau)}}{\sum_{k=1}^{n_c^t} \exp{(\langle \boldsymbol{f}, \boldsymbol{c}_k \rangle / \tau)} + \sum_{k=1}^{n_o^t} \exp{(\langle \boldsymbol{f}, \boldsymbol{v}_k \rangle / \tau)}}$$

      **3:** Update instance features $\{\boldsymbol{v}\}$ in the hybrid memory with $\{\boldsymbol{f}_i^t\}$ and momentum $m^t$ (Eq. (4));
      **4:** Update cluster centroids $\{\boldsymbol{c}\}$ with updated $\{\boldsymbol{v}\}$ in the hybrid memory (Eq. (2));
    **end for**
  **end for**

---

## B    More Discussions

**Comparison with ECN [62, 63].**    There is an existing work, ECN [62] with its extension version [63], which also adopts a feature memory for the domain adaptive person re-ID task. Comparison results in Table 2 demonstrate the superiority of our proposed method, and there are three main

differences between our method and ECN. (1) Our proposed hybrid memory dynamically provides all the source-domain class-level, target-domain cluster-level and un-clustered instance-level supervisory signals, while the memory used in ECN only provides instance-level supervisions on the target domain. (2) We use unified training of source classes, target clusters and target outliers, while ECN uses multi-task learning and treats source and target classes separately. (3) We propose a self-paced learning strategy to gradually refine the learning targets on both clusters and un-clustered instances, while ECN adopts noisy $k$-nearest neighbors as learning targets for all the samples without consideration of uneven density in the latent space.

## C  More Implementation Details

We implement our framework in PyTorch [35] and adopt 4 GTX-1080TI GPUs for training[†]. The domain adaptation task with both source-domain and target-domain data takes $\sim 3$ hours for training, and the unsupervised learning task with only target-domain data takes $\sim 2$ hours for training on Market-1501 and PersonX datasets. When training on MSMT17, VehicleID, VeRi-776 and VehicleX datasets, time needs to be doubled due to over $2\times$ images in the training set.

### C.1  Network Optimization

We adopt an ImageNet [7]-pretrained ResNet-50 [18] up to the global average pooling layer, followed by a 1D BatchNorm layer and an $L_2$-normalization layer, as the backbone for the encoder $f_\theta$. Domain-specific BNs [3] are used in $f_\theta$ for narrowing domain gaps. Adam optimizer is adopted to optimize $f_\theta$ with a weight decay of 0.0005. The initial learning rate is set to 0.00035 and is decreased to 1/10 of its previous value every 20 epochs in the total 50 epochs. The temperature $\tau$ in Eq. (1) is empirically set as 0.05. The hybrid memory is initialized by extracting the whole training set with the ImageNet-pretrained encoder $f_\theta$, and is then dynamically updated with $m^s = m^t = 0.2$ in Eq. (3)&(4) at each iteration.

### C.2  Training Data Organization

During training, each mini-batch contains 64 source-domain images of 16 ground-truth classes (4 images for each class) and 64 target-domain images of *at least* 16 pseudo classes, where target-domain clusters and un-clustered instances are all treated as independent pseudo classes (4 images for each cluster or 1 image for each un-clustered instance). The person images are resized to $256 \times 128$ and the vehicle images are resized to $224 \times 224$. Random data augmentation is applied to each image before it is fed into the network, including randomly flipping, cropping and erasing [61].

### C.3  Target-domain Clustering

Following the clustering-based UDA methods [11, 10, 38], we use DBSCAN [9] and Jaccard distance [60] with $k$-reciprocal nearest neighbors for clustering before each epoch, where $k = 30$. For DBSCAN, the maximum distance between neighbors is set as $d = 0.6$ and the minimal number of neighbors for a dense point is set as $4$. In our proposed self-paced learning strategy described in Section 3.2, we tune the value of $d$ to loosen or tighten the clustering criterion. Specifically, we adopt $d = 0.62$ to form the looser criterion and $d = 0.58$ for the tighter criterion, denoted as $\Delta d = 0.02$. The constant threshold $\alpha$ for identifying independent clusters is defined by the top-90% $\mathcal{R}_{\text{indep}}$ before the first epoch and remains the same for all the training process. The dynamic threshold $\beta$ for identifying compact clusters is defined by the maximum $\mathcal{R}_{\text{comp}}$ in each cluster on-the-fly, *i.e.*, we preserve the most compact points in each cluster.

## D  Additional Experimental Results

### D.1  Performance with IBN-ResNet [34]

Instance-batch normalization (IBN) [34] has been proved effective in object re-ID methods in either unsupervised [11] or supervised [30] learning tasks. We evaluate our framework with IBN-ResNet as

---

[†]https://github.com/yxgeee/SpCL

Table 6: Comparison of different backbones in our framework, *i.e.*, ResNet-50 and IBN-ResNet.

| Source | Target | Ours *w/* ResNet-50 | | | | Ours *w/* IBN-ResNet | | | |
|---|---|---|---|---|---|---|---|---|---|
| | | mAP | top-1 | top-5 | top-10 | mAP | top-1 | top-5 | top-10 |
| Market-1501 | MSMT17 | 26.8 | 53.7 | 65.0 | 69.8 | **31.0** | **58.1** | **69.6** | **74.1** |
| MSMT17 | Market-1501 | 77.5 | 89.7 | 96.1 | 97.6 | **79.9** | **92.0** | **97.1** | **98.1** |
| PersonX | Market-1501 | 73.8 | 88.0 | 95.3 | 96.9 | **77.9** | **90.5** | **96.1** | **97.7** |
| PersonX | MSMT17 | 22.7 | 47.7 | 60.0 | 65.5 | **25.4** | **50.6** | **63.3** | **68.3** |
| VehicleID | VeRi-776 | **38.9** | **80.4** | **86.8** | **89.6** | 38.0 | 79.7 | 85.8 | 88.4 |
| VehicleX | VeRi-776 | **38.9** | **81.3** | **87.3** | **90.0** | 37.8 | 80.7 | 86.1 | 89.2 |
| None | Market-1501 | 73.1 | 88.1 | 95.1 | 97.0 | **73.8** | **88.4** | **95.3** | **97.3** |
| None | MSMT17 | 19.1 | 42.3 | 55.6 | 61.2 | **24.0** | **48.9** | **61.8** | **67.1** |
| None | VeRi-776 | **36.9** | **79.9** | **86.8** | **89.9** | 36.6 | 79.1 | 85.9 | 89.2 |

the backbone of the encoder, which is formed by replacing all BN layers in ResNet-50 [18] with IBN layers. As shown in Table 6, the performance can be further improved with IBN-ResNet except for the vehicle datasets.

## D.2 Self-paced Learning Strategy on Other Clustering Algorithms

Table 7: Evaluate our framework over Agglomerative Clustering [1] algorithm. Experiments are conducted on the tasks of unsupervised person re-ID.

| Clustering | Market-1501 | | | |
|---|---|---|---|---|
| | mAP | top-1 | top-5 | top-10 |
| Agglomerative Clustering *w/o* self-paced strategy | 70.4 | 87.1 | 94.7 | 96.6 |
| Agglomerative Clustering *w/* self-paced strategy | **75.2** | **89.7** | **95.8** | **97.5** |

In order to verify that our proposed self-paced learning strategy with cluster reliable criterion is still effective when creating pseudo labels with other clustering algorithms, we conduct experiments by replacing the original DBSCAN algorithm with Agglomerative Clustering [1] algorithm. As shown in Table 7, significant 4.8% mAP improvements can be observed when applying the self-paced learning strategy. What is interesting is that the final performance is even better than that on DBSCAN.

## D.3 Cluster Reliable Criterion *v.s.* HDBSCAN [2]

Table 8: Comparison between DBSCAN *w/* our cluster reliable criterion and HDBSCAN [2]. Experiments are conducted on the tasks of unsupervised person re-ID.

| Clustering | Market-1501 | | | | MSMT17 | | | |
|---|---|---|---|---|---|---|---|---|
| | mAP | top-1 | top-5 | top-10 | mAP | top-1 | top-5 | top-10 |
| DBSCAN *w/* our cluster reliable criterion | **73.1** | **88.1** | **95.1** | **97.0** | **19.1** | **42.3** | **55.6** | **61.2** |
| HDBSCAN | 71.7 | 87.7 | 95.0 | 96.3 | 15.7 | 39.2 | 51.3 | 56.7 |

The intuition of our cluster reliable criterion is to measure the stability of clusters by hierarchical structures, which shows similar motivation as HDBSCAN [2]. So we test HDBSCAN to replace our reliability criterion and observe 1.4%/3.4% mAP drops on unsupervised Market-1501/MSMT17 tasks (Table 8), which indicates that DBSCAN with our cluster reliability criterion is more suitable than HDBSCAN in the proposed framework.

# E  Parameter Analysis

We tune the hyper-parameters on the task of MSMT17→Market-1501, and the chosen hyper-parameters are directly applied to all the other tasks.

## E.1 Temperature $\tau$ for Contrastive Loss

As demonstrated in Figure 4, our framework achieves the optimal performance when setting the temperature $\tau$ as 0.05 in Eq. (1) on the task of MSMT17→Market-1501. One may find that the performance varies with different values of $\tau$, but note that all methods using temperature contrastive

Figure 4: Performance of our framework with different values of temperature $\tau$.

function (*e.g.*, [62, 63, 48, 17, 4, 33]) have similar effects on $\tau$. We set $\tau = 0.05$ following [62, 63] and achieve the best performance using the same $\tau = 0.05$ for 6 UDA tasks (Table 2) and 3 unsupervised tasks (Table 4), showing the robustness of $\tau =$ fixed 0.05.

## E.2 Momentum Coefficients $m^s, m^t$ for Hybrid Memory

Figure 5: Performance of our framework with different values of $m^t$ when $m^s = 0.2$.

Figure 6: Performance of our framework with different values of $m^s$ when $m^t = 0.2$.

Figure 7: Performance of our framework with different values of $m^s, m^t$ when $m^s = m^t$.

Our proposed hybrid memory simultaneously stores and updates the source-domain class centroids with momentum $m^s$ in Eq. (3) and the target-domain instance features with momentum $m^t$ in Eq. (4). We adopt $m^s = m^t = 0.2$ in our experiments by tuning such hyper-parameter on the task of MSMT17→Market-1501.

We find that the value of $m^t$ is critical to the optimal performance (Figure 5) while our framework is not sensitive to the value of $m^s$ (Figure 6), so we adopt the same momentum coefficient on two domains for convenience, $i.e.$, $m^s = m^t$. Despite the value of $m^t$ affects the final performance, the results of our framework are robust when $m^t$ changes within a large range, $i.e.$, $[0.2, 0.6]$ in Figure 7.

### E.3  Residual $\Delta d$ for Cluster Reliability Criterion

Figure 8: Performance of our framework with different values of $\Delta d$ in the cluster reliability criterion.

As described in Section C.3, we tune the value of the maximum neighbor distance $d$ with a residual $\Delta d = 0.02$ to measure the cluster reliability in our self-paced learning strategy. As shown in Figure 8, $\Delta d = 0.00$ can be thought of as removing the self-paced strategy from training, which is the same as "Ours $w/o$ $\mathcal{R}_{\mathrm{comp}} \& \mathcal{R}_{\mathrm{indep}}$" in Table 5. Our method could achieve similar performance when $\Delta d$ changes within $[0.02, 0.05]$, which indicates that our proposed reliability criterion is not sensitive to the hyper-parameter $\Delta d$.

## Footnotes

*Dapeng Chen is the corresponding author.