[Reviews · NeurIPS 2020]

Review 1

Summary and Contributions: This paper proposed a self-paced contrastive learning framework and utilise hybrid memory to jointly distinguishes source-domain classes, and target-domain clusters and un-clustered instances. The significant improvement of the proposed method over the baselines are shown on multiple benchmarks.

Strengths: 1. The paper proposes an innovative way to fully utilize all data during domain adaptation for ReID, while other methods discard source-domain knowledge and target-domain outliers. 2. To include different sources for training, the author defines a unified contrastive loss to jointly consider three sources of supervision, with appropriate adaptation on source-domain to match semantics. 3. The hybrid memory design provides centroids/instance for the unified contrastive loss, while live update for source-domain and self-paced learning for clustering in target-domain refresh the memory. 4. The self-paced learning mechanism helps forming more reliable cluster centroids by introducing two metrics, independence and compactness, to make the clustering process self-adaptive. 5. Abundant studies show the effectiveness of the designed components, together with an oracle setup to reveal a possible upper bound.

Weaknesses: 1. The poor result of reimplemented MoCo in 4.3 Table 4 needs further explanation and reasoning to account for. 2. Need clearer description on difference between the setup of ‘Src. Class + tgt. Cluster (w/ self-paced)’ in ablation study Table 5 and the full model. If the only difference is target instance, where does the ~10 difference in mAP come from?

Correctness: The claims and method are correct. No doubt for empirical methodology.

Clarity: Most part of this paper is easy to follow and clear for practice.

Relation to Prior Work: Clearly discussed in related work part.

Reproducibility: Yes

Additional Feedback: Comments after the rebuttal ------------------------------------------------------------ I agree with the concerns brought up by R6 and R9 that the paper shares similar ideas with memory banks and momentum contrast. Thus, I decrease the score to "marginally below the acceptance threshold".


Review 2

Summary and Contributions: This paper proposes a self-paced contrastive learning framework for object re-identification. They learn feature representations by contrasting each feature against the source classes, target clusters (unsupervised), and remaining target samples (which kind of act as a cluster with a single member). They also use a memory bank to keep prototype representations for each source class, target cluster and target outlier. The update of these prototypes are performed through momentum updates. They claim the state of the art results in object re-identification benchmarks.

Strengths: + Smoothly combines the ideas of [13] and [45] in a unified framework specifically targeting the object re-identification problem. + They present the state of the art results in several object re-identification benchmarks. + The authors provide a decent ablation of their components.

Weaknesses: - As system keeps features for all the instances in the target domain in a memory bank, there scale well with large number of unlabelled instances. - Cluster reliability measures are a bit adhoc and should be explained better. Also it doesn't seem to have too large effect on the final results.

Correctness: Appears to be so.

Clarity: It is in a decent form but presentation can be improved. There are many repetitions.

Relation to Prior Work: Paper shares similar ideas with [45] (instance discrimination with memory banks) and [13] (MOCO - Momentum contrast), though the proposed method is sufficiently different from both papers and operates on an entirely different problem (object re-id). That said, there is a value in acknowledging and explaining these relations. For instance the loss for target outliers is quite similar to instance discrimination, the main difference is having also the class and cluster centroids. Keeping all these prototypes in a hybrid memory also resembles the memory bank of [A]. The way the prototypes and features updated also has loose similarities with momentum contrast idea. Establishing these links would increase the quality and readability of the paper. [45] Unsupervised Feature Learning via Non-Parametric Instance-level Discrimination Zhirong Wu, Yuanjun Xiong, Stella Yu, Dahua Lin [13] Momentum Contrast for Unsupervised Visual Representation Learning Kaiming He, Haoqi Fan, Yuxin Wu, Saining Xie, Ross Girshick

Reproducibility: No

Additional Feedback: Update after the rebuttal: After reading the other reviews and the rebuttal, I'm a bit concerned about the ethical issues. I wasn't aware of it before but the general ethical concern on DukeMTMC dataset and Duke University's removal should be a good enough reason to not to report on this dataset in any scientific paper. The fact that this is overlooked and not even mentioned in the ethics and broader impact section raises serious concerns about this paper. Also accepting the paper in its current form would be encouraging the future usage of this dataset in follow-up papers. In the light of these concerns on ethics, I would suggest removing the results on DukeMTMC following Duke University's (the publisher of the dataset) decision. Unfortunately that would make the paper a bit weaker than it currently is.


Review 3

Summary and Contributions: The paper addresses the problem of unsupervised domain adaptation (UDA) with a strong emphasis on the task of re-identification. The contributions include: the extension of the contrastive softmax with specific selection of class centroids from the labeled source domain along with cluster centroids and outlier instances of the target domain, a hybrid memory which can be considered as a non-parametric model used to maintain and update the state of clusters and outliers between epochs. A form of self-paced learning is proposed which considers cluster reliability of pseudo labels to remove difficult/noisy samples from the target domain each epoch for smoother and more reliable training of domain transfer. Predominantly an empirical paper without theoretical novelty. That said the novelty combination of pseudo label clustering and network training within a self-paced framework can be considered as a novel contribution that leads the proposed method to outstanding results.

Strengths: The paper is written reasonably well with clear structure and presentation. An extensive evaluation is given for several domain transfer scenarios across seven different datasets. In all experiments the proposed method significantly outperforms other UDA methods. The authors have provided links to their code. Despite not yet having run the code to check for reproducibility the code appears to be complete (also see DukeMTMC note in weaknesses). The UDA-ReID problem is interesting from the perspective of learning representations, generalising across domains and towards unsupervised learning.

Weaknesses: I wasn't able to download the DukeMTMC dataset used in most experiments. The link http://vision.cs.duke.edu/DukeMTMC in the reference from the provided source. It would appear that this dataset has been taken down by Duke university due to a potential privacy infringement since June 2019. This would make it difficult to ethically reproduce many of the experiments of this paper. There are enough other datasets used in this work to validate the proposed method without DukeMTMC. Given that DukeMTMC has been taken down so long ago I do not know what it is used in the experiments. Especially for Tables 3, 4, and 5 where MSMT17 could be used instead. For a NeurIPS paper I would expect more of a theoretical grounding for the various design choices. However, the paper does motive several choices using common intuition e.g. measuring cluster reliability due to the uneven density in the latent space.

Correctness: No novel or strong theoretical claims are made that require a proof. The empirical methodology uses common metrics and appear consistent with comparative papers.

Clarity: Overall the paper is well written with a clear and illustrative description of the proposed method. Some grammatical and clarity issues remain and should be addressed. Line 145 should read ", the performance drops significantly." "the target-domain instance features {v} are only initialized once at the beginning of the whole learning algorithm" on line 165 is contradicted in lines 166, 179 and with equation 4.

Relation to Prior Work: While the appendix attempts to highlight the differences between the Hybrid memory and the memory used in ECN, more could be done to express the differences in the Hybrid memory updating procedure and the MoCo[13] method. Other than the use of centroids in a non-parametric fashion there appears to be significant similarity which should be acknowledged in section 3.1.2 While this paper differs in its use or application of measuring cluster stability for self-paced learning, it also overlooks some earlier work in cluster stability analysis. Namely, more could be done to motivate the use of thresholding cluster compatness and independence using selected hyper-parameters (e.g. density and alpha beta) with other measures of dissimilarity inn heirarchical clustering (e.g. [A]) or forms of consensus clustering [B]. [A] R.J.G.B. Campello, D. Moulavi, A. Zimek and J. Sander (2015) "Hierarchical Density Estimates for Data Clustering, Visualization, and Outlier Detection", ACM Trans. on Knowledge Discovery from Data [B] A. Strehl, J. Ghosh, (2002). "Cluster ensembles – a knowledge reuse framework for combining multiple partitions" Journal on Machine Learning Research

Reproducibility: Yes

Additional Feedback: More discussion on the sensitivity of hyper parameters and choice of clustering algorithm should be made with regard to their effect on the UDA problem. While experiments with DukeMTMC are valuable for comparing with prior work, it is important to respect the privacy of the people in the dataset and acknowledge that this dataset is no longer available. More emphasis should be placed on the other experiments with publicly available datasets. Overall the paper is very strong on the empirical side with very impressive results. Update: In the rebuttal the authors continue to use Duke as an empirical datum in responses to other reviewers and have not acknowledged that this dataset has officially been decommissioned, meaning that other researchers cannot access this dataset for reproducibility and more importantly that any storage or distribution of said dataset is considered unethical as it is known to breach privacy standards. See: https://www.dukechronicle.com/article/2019/06/duke-university-facial-recognition-data-set-study-surveillance-video-students-china-uyghur The act of disregarding this takedown notice stands against the comments in the Broader Impact section around infringement of people's privacy. While this dataset is common for ReID it has been removed for over 12 months and the camera-ready version of this paper should acknowledge this and as suggested remove or reduce the use of Duke. The additional results support the effectiveness of this approach and I'm sure it a revised version of the paper would do just as well on other datasets as shown but this would require significant changes.


Review 4

Summary and Contributions: This work address the task of unsupervised domain adaptation for object re-ID. It proposes to use a contrastive learning framework with source-domain class-level, target-domain cluster-level and target-domain instance-level supervision. It also defines two criteria of independence and compactness to help obtain reliable clusters for learning. Experiments are conducted on person and vehicle re-ID and some ablation studies are also presented.

Strengths: + The task of unsupervised domain adaptation is interesting and challenging. + Multiple datasets are used for evaluations. + Related works are appropriately discussed and compared.

Weaknesses: - The main idea of this method is unified contrastive learning. However, the strategy of joint learning of source and target domain is not new although different methods implement with different losses (e.g., in [57,58]). It is also natural that the performance on source domain with joint learning of source and target domains is higher than finetuing with target data only. Besides, the form of non-parametric contrastive learning is widely used in general unsupervised visual representation learning methods (such as MoCo and SimCLR) and is not new in this method. - The assumption of the proposed unified contrastive learning is that the source domain has disjoint classes with target domain as it needs to collect cross-domain samples as negatives. It may meet with the current UDA benchmarks but the generality of this method based on such assumption is limited in those real-world practical application scenarios where no prior knowledge are available on target data. Existing methods which optimize source and target domains separately thus show more advantages in this aspect. - It is not clear why optimizing class-level and instance-level contrastive losses simultaneously will work. Class-level supervision is different with instance-level supervision as optimization target. The experiments of MoCo on UDA do not work, which also implies that instance-level supervision is not suitable for distinguishing semantic classes in the object re-ID tasks. It lacks sufficient explanations and corresponding ablation studies to clarify this point. It is hard to convince me why such contrastive loss can work with current content and experiments. - The ablation studies are not clear and sufficient enough. (1) What are the differences between "src class + tgt class (w/o self-paced)" and "ours w/o self-paced r_comp & r_indep"? It is not clear which algorithmic components self-paced learning contains and it lacks necessary detailed descriptions of the setting of these experiments. (2) The ablation studies of different combinations of class-level, cluster-level and instance-level are not presented. Since the unified contrastive learning is the core idea of this method, these experiments are necessary but missing unfortunately. (3) I'm also confused with the differences between w/o self-paced and Delta_d=0. (4) Why did using learnable classifiers perform worst than using class centroids for source domain? It also lacks necessary theoretical analysis and explanations. - The strategies of independence and compactness of clusters seem to be tricky and incremental. The strategies need multiple manual parameters based on DBSCAN clustering. From Table 5, on Market-to-Duke task, only 0.8% mAP drops w/o r_indep and only 1.3% mAP drops w/o r_comp. The results implies incremental contributions of such strategies. - In the unified contrastive learning (Eq. 1), if f is a target-domain un-clustered outlier, it is not clear how to collect its corresponding positive samples. - Softmax-based losses and triplet losses are widely used in object re-ID tasks. It is necessary to compare them with the contrasitve loss. But the comparisons and analysis are missing in this work. - The parameter analysis experiments show that tuning temperature param has a large impact on final re-ID performance, e.g., 68.8% with 0.05 vs 57.4% with 0.09. Such large gap (11.4% mAP) is even higher than other major algorithmic components. It may imply that this method is sensitive to this param and not robust enough to extend to other tasks. It also raises the concerns that whether the improvement mainly comes from hyper-parameters tuning. - In Figure 3, the metric of cluster number is not good enough to show the quality of clustering. A better way is to use some quantitative metrics (e.g., NMI or F-measure) to check how good/bad clusters a method obtains.

Correctness: It needs more clarifications and experiments on method design, e.g., why combining instance-level and class-level supervision can work?

Clarity: The sentences are well written. But it lacks some necessary ablation experiments and analysis.

Relation to Prior Work: It lacks some necessary ablation stuides and detailed descriptions of some experimental settings.

Reproducibility: Yes

Additional Feedback: See above. Update after the rebuttal: I have read the other reviews as well as the rebuttal. The rebuttal addressed part of the raised issues. However, I still have concerns on the technical novelty, theoretical basis and ethical problem. There are about 80% of experiments that rely on the questionable Duke dataset. I think this paper needs major improvement and thus keep my original rating.

[Author Response · NeurIPS 2020]

**Q1:** Explain the poor results of MoCo in Tab. 4. **A1:** MoCo alone underperforms because it treats each **instance** as a single class, while the core of re-ID tasks is to encode and model **intra-/inter-class** variations. MoCo is good at unsupervised pre-training but its resulting networks need finetuning with (pseudo) class labels.

**Q2:** "Src. class+tgt. cluster (w/ self-paced)" v.s. "Ours (full)" in Tab. 5. **A2:** The difference is whether using un-clustered outliers. Reasons for the drop: 1) There are many un-clustered outliers ($>$ half of all samples), especially in early epochs. 2) Outliers serve as difficult samples and excluding them over-simplifies the training task. 3) The baseline doesn't update outliers in the memory, making them unsuitable to be used in pseudo classes in the later epochs.

 **Q3:** Hard to scale up? **A3:** Caching a 2048d instance needs $\sim 0.05$M. Our method can cache 10,000,000+ instances in 500G CPU memory. If caching in 11G GPU memory, 200,000+ instances can be easily stored.

**Q4:** Explain the cluster reliability criterion better. **A4:** The intuition is to measure the stability of clusters by hierarchical structures, *i.e.*, a reliable cluster should be consistent in clusters at multiple levels. It leads to evident performance gains, *i.e.*, $>2\%$ mAP gains on two tasks in Tab. 5 ("Ours w/o self-paced $\mathcal{R}_{comp}\&\mathcal{R}_{indep}$" v.s. "Ours (full)").

**Q5:** Relations to [13, 45]. **A5:** We discussed the differences from [13, 45] on L3-9 of supplementary material and we will further discuss their relations to our work following your advice.

 **Q6:** DukeMTMC is not available. **A6:** We added experiments on MSMT17 as suggested. For the source-domain performance on Market (Tab. 3), our method can boost the mAP by +6.3% by training with unsupervised MSMT. For the unsupervised performance (Tab. 4), we reached 19.1% mAP, outperforming 11.2% mAP of SOTA [42].

**Q7:** Lack of theoretical grounding. **A7:** Indeed, the effectiveness is mainly demonstrated via ablation studies in both main text and supplementary, which show significant improvements. We will look into more theories in future studies.

**Q8:** Difference to memory usage in MoCo [13]. **A8:** Other than centroids, we for the first time treat clusters and instances as equal classes. Our self-paced strategy dynamically determines confident clusters and un-clustered instances.

**Q9:** Relation to [A, B]. **A9:** We tested HDBSCAN [A] to replace our reliability criterion and observed 0.9%/4.3% mAP drops on unsupervised Market/MSMT tasks. We will further discuss earlier works and improve our method.

**Q10:** Hyper-parameter sensitivity and choice of clustering algorithms. **A10:** We discussed hyper-parameters in Sec. E of Appendix. We adopted DBSCAN to fairly compare with [9, 50, 47, 51] in Tab. 2. We also tested Agglomerative Clustering algorithm on unsupervised Market: 74.9% mAP by "Ours (full)" v.s. 70.4% mAP by "Ours w/o self-paced".

 **Q11:** Joint learning is not new [57, 58]. The gain is natural. **A11:** We use unified training of source classes, target clusters and target outliers, which is totally different from [57, 58]. They use multi-task learning and treat source and target class *separately* (Appendix L10-20). Naive cross-domain training would hurt the performance [10].

**Q12:** The form of contrastive learning is not new. **A12:** We **never** claimed that the form of contrastive learning is our novelty. We focused on exploiting all available information by jointly distinguishing different kinds of prototypes with a novel hybrid memory. We discussed the differences from previous contrastive learning methods on L92-98 (main paper) and L3-9 (Appendix). Previous methods (*e.g.*, MoCo) fail in Tab. 4. See **A1** for reasons.

**Q13:** The assumption of disjoint label sets is unrealistic. **A13:** Actually quite common in real-world cases. One collect annotations from city A and generalize the models to other cities. Face recognition datasets have similar phenomenon.

**Q14:** Why simultaneous class- and instance-level loss work? **A14:** MoCo *alone* not working on re-ID tasks doesn't imply that the proposed joint class+cluster+instance training would fail. Cluster outliers are crucial to the training (see **A2**), and treating them as single-instance classes boosts the performance significantly, given the ablation study in Tab. 5: using source class-level + only target instance-level losses ("Src. class+tgt. instance") totally fails, similar to MoCo; using source class-level + only target cluster-level losses ("Src. class+tgt. cluster (w/ self-paced)") shows inferior result.

**Q15:** Lack of ablation studies. **A15:** 1) "Src. class + tgt. cluster (w/o self-paced)" discards both self-paced strategy (cluster reliable criterion) and un-clustered instances from training. "Ours w/o self-paced $\mathcal{R}_{comp}\&\mathcal{R}_{indep}$" only removes self-paced strategy. 2) All the combinations of losses have been investigated in Tab. 5, *i.e.*, "Src. class", "Src. class + tgt. instance" and "Src. class + tgt. cluster". "tgt. cluster + tgt. instance" is the same as "Ours w/o source-domain data" in Tab. 4. 3) Same, as described on L79-80 of Appendix. 4) The learnable classifiers in the source domain don't match the semantic meaning of target-domain centroids and thus cause inferior performance (L142-144).

**Q16:** Reliability criterion is tricky and incremental. **A16:** It is meaningless to evaluate $\mathcal{R}_{comp}, \mathcal{R}_{indep}$ independently, as they complement each other and leads to over 2% mAP gain. Please see also **A4** for intuition.

**Q17:** Positive sample for un-clustered outlier $f_k$. **A17:** It is $v_k$ (L139-140) cached in the hybrid memory (Eq. (4)).

**Q18:** Compare to softmax/triplet loss. **A18:** Duke$\rightarrow$Market (mAP): 25.0% by cross-entropy loss, 30.1% by cross-entropy+triplet loss, 74.2% by unified contrastive+triplet loss, which are all lower than those reported (76.7%). As both cross-entropy and unified contrastive loss are variants of softmax loss, the key to success is our well-designed hybrid memory, which provides **continuous** learning targets for **dynamically** changing clusters and un-clustered instances.

**Q19:** The Temperature $\tau$ is sensitive. **A19:** All methods using temperature softmax function (e.g. [57, 58]) have similar effects on $\tau$. See also Tab. 1 of [57, 58]. We set $\tau = 0.05$ following [57, 58] and achieve the best performance using the same $\tau = 0.05$ for 8 UDA tasks (Tab. 2) and 2 unsupervised tasks (Tab. 4), showing the robustness of $\tau$=fixed 0.05.

**Q20:** Evaluate the clusters. **A20:** At the last epoch of Duke$\rightarrow$Market, F1 & NMI scores are: 0.82 & 0.94 (Ours full), 0.79 & 0.92 (Ours w/o self-paced), 0.73 & 0.90 (Src. class + tgt. cluster (w/ self-paced)). We will show the curves.

[Meta-Review · NeurIPS 2020]

Three of the four reviewers originally recommended marginal accept or accept (7, 6, 6) as they felt the paper provided a good empirical contribution to the field of adaptive re-identification and its results were strong. R9 was more negative and had concerns around experiments. One reviewer pointed out that the DukeMTMC extensively used in the paper has been taken down 12 months ago and its use should be discontinued. Because of the ethical concerns around this, the paper underwent additional review by the ethics panel, which recommended that the dataset should NOT be used in an accepted NeurIPS paper. Some excerpts from the ethics reviewers are below: -- "... the dataset collection involved non-consensual video surveillance of students on Duke University campus. It is unlikely that all students even knew they were being recorded, and their relative lack of power with respect to the institution surveilling them also raises concerns about the ability to meaningfully object to the surveillance." -- "Including the dataset in the paper as-is would be problematic, as it would contribute to this mainstream use of the dataset. Referencing the issues and discouraging future use of the dataset would help mitigate this, as would full removal of the results." -- "The fact that others use the dataset uncritically does not make it appropriate, but it could have contributed to the authors being unaware of the issue, and that awareness may vary geographically." -- "The Broader Impact section should state more clearly that surveillance is the typical goal of re-id systems. It should also state the re-id systems often rely on non-consensual surveillance data for their training. The Broader section should also state that the demographic makeup of the datasets used are not representative of the broader population." -- "...this innovation ... can potential equip malicious actors with the ability of actors to surveil person(s) or groups through multiple cctv cameras without their consent. For potential mitigations, I would strongly urge the authors at a minimum to discuss the potential harms of person re-id due to surveillance, explain the consent challenges with MTMC data collection, as well as remove the Duke MTMC dataset from their evaluation." As a result of the above ethics reviews, the initially positive reviewers have downgraded to marginally below accept, citing concerns around the use of the dataset. The AC read the reviews, rebuttal and ethics reviews and agrees that the paper should remove all experiments on the Duke dataset. However the AC does not think it is fair to reject the paper based on its use of the dataset alone, as its takedown is relatively recent and the authors pointed out that it appears in CVPR 2020 papers, necessitating a comparison. The authors have promised to take the dataset out of the camera ready and replace with experiments on other datasets, where they also promise that similarly strong results can be achieved. The AC therefore recommends a **conditional accept** (a rare and exceptional case reserved only for papers with ethical review), which means the paper is accepted on the condition that the final version 1) removes all results on DukeMTMC, 2) makes it clear that the dataset has been taken down and should no longer be used, and 3) reproduces all technical contributions and results of the original submission using other datasets. The authors are also encouraged to take into account other reviewer suggestions in preparing the camera ready. The AC discussed this decision with the SAC and PCs. ******************************* Note from Program Chairs: The camera-ready version of this paper has been reviewed with regard to the conditions listed above, and this paper is now fully accepted for publication.